# Comparison of the Immune Responses to COVID-19 Vaccines in Bangladeshi Population

**DOI:** 10.3390/vaccines10091498

**Published:** 2022-09-08

**Authors:** Protim Sarker, Evana Akhtar, Rakib Ullah Kuddusi, Mohammed Mamun Alam, Md. Ahsanul Haq, Md. Biplob Hosen, Bikash Chandra Chanda, Farjana Haque, Muntasir Alam, Abdur Razzaque, Mustafizur Rahman, Faruque Ahmed, Md. Golam Kibria, Mohammed Zahirul Islam, Shehlina Ahmed, Rubhana Raqib

**Affiliations:** 1Infectious Diseases Division, International Centre for Diarrhoeal Disease Research (icddr,b), Dhaka 1212, Bangladesh; 2Laboratory Sciences and Services Division, International Centre for Diarrhoeal Disease Research (icddr,b), Dhaka 1212, Bangladesh; 3Health Systems and Population Studies Division, International Centre for Diarrhoeal Disease Research (icddr,b), Dhaka 1212, Bangladesh; 4Sheikh Russel Gastroliver Institute & Hospital, Dhaka 1212, Bangladesh; 5Embassy of Sweden in Dhaka, Dhaka 1212, Bangladesh; 6Foreign Commonwealth & Development Office (Bangladesh), Dhaka 1212, Bangladesh

**Keywords:** B cell, T cell, cellular immunity, humoral immunity, neutralizing antibodies, SARS-CoV-2

## Abstract

Background: The adaptive immune response is a crucial component of the protective immunity against SARS-CoV-2, generated after infection or vaccination. Methods: We studied antibody titers, neutralizing antibodies and cellular immune responses to four different COVID-19 vaccines, namely Pfizer-BioNTech, Moderna Spikevax, AstraZeneca and Sinopharm vaccines in the Bangladeshi population (*n* = 1780). Results: mRNA vaccines Moderna (14,655 ± 11.3) and Pfizer (13,772 ± 11.5) elicited significantly higher anti-Spike (S) antibody titers compared to the Adenovector vaccine AstraZeneca (2443 ± 12.8) and inactivated vaccine Sinopharm (1150 ± 11.2). SARS-CoV-2-specific neutralizing antibodies as well as IFN-γ-secreting lymphocytes were more abundant in Pfizer and Moderna vaccine recipients compared to AstraZeneca and Sinopharm vaccine recipients. Participants previously infected with SARS-CoV-2 exhibited higher post-vaccine immune responses (S-specific and neutralizing antibodies, IFN-γ-secreting cells) compared to uninfected participants. Memory B (B_MEM_), total CD8^+^T, CD4^+^ central memory (CD4^+^_CM_) and T-regulatory (T_REG_) cells were more numerous in AstraZeneca vaccine recipients compared to other vaccine recipients. Plasmablasts, B-regulatory (B_REG_) and CD4^+^ effector (CD4^+^_EFF_) cells were more numerous in mRNA vaccine recipients. Conclusions: mRNA vaccines generated a higher antibody response, while a differential cellular response was observed for different vaccine types, suggesting that both cellular and humoral responses are important in immune monitoring of different types of vaccines.

## 1. Introduction

Growing evidence suggests that the adaptive immune response is critical for the development of protective immunity against SARS-CoV-2, including viral clearance and the persistence of antiviral immunity [1]. Generation of neutralizing antibodies that specifically target the receptor-binding domain (RBD) of the spike (S) protein is considered to be essential in controlling SARS-CoV-2 infection [2,3]. A robust adaptive immune response with the presence of RBD and S-specific neutralizing antibodies, memory B cells, and T-cell response has been found in patients who have recovered from infection [4,5,6,7]. Although circulating antibodies derived from plasma cells wane over time, long-lived immunological memory can persist in expanded clones of memory B cells [5,7].

The world responded rapidly by developing, evaluating and deploying multiple COVID-19 vaccines, with 69% of the world population having received at least one dose of a COVID-19 vaccine [8]. About 12.4 billion doses of COVID-19 vaccines have been administered globally. Eight COVID-19 vaccines have so far been approved by the drug-regulatory authority for use in Bangladesh. The Pfizer-BioNTech Comirnaty (BNT162b2) and Moderna Spikevax (mRNA-1273) use spike-encoding mRNA in lipid nanoparticles. AstraZeneca (ChAdOx1-S/Covishield), Johnson & Johnson/Janssen (Ad26.COV2.S) and Gamaleya (Sputnik V) are adenovector vaccines. Novavax/COVOVAX (NVX-CoV2373) is a protein subunit vaccine. All these vaccines use spike protein of the first emerged SARS-CoV-2 in Wuhan, China, as the target immunogen to be presented in native conformation for inducing high levels of neutralizing antibodies. The Sinopharm (BBIBP-CorV) and Sinovac (CoronaVac) are inactivated whole-virus vaccines that contain diverse viral proteins with theoretical potential to broaden immune protection against the variants of concern (VOCs) beyond the spike-protein-specific immune response. Many of these vaccines have been demonstrated to be successful in reducing severity of infections, hospitalization and deaths [9,10,11,12,13,14,15]. Induction of neutralizing antibodies [16,17] and cellular immunity [18,19,20,21,22] post-vaccination is likely to play an important role in the protective immunity. Limited studies have compared humoral and/or cellular responses developed by some of these COVID-19 vaccines showing variations in the generated immune responses [20,23,24,25,26,27]. There are no reports yet on the comparative immune responses to different COVID-19 vaccines in Bangladesh. Here, we aimed to study antibody titers, neutralizing capacities of antibodies and cellular immune response after completion of two doses of COVID-19 vaccines in the Bangladeshi population.

## 2. Materials and Methods

### 2.1. Study Design and Participant Recruitment

Participants (*n*-1780, age range 17 to 88 years) were enrolled within 2–4 weeks after receiving two doses of COVID-19 vaccines from the Sheikh Russel Gastroliver Institute & Hospital (SRGIH) as well as from the urban communities of 5 major divisions from November 2021 to May 2022. The vaccinees included in this study received one of the four vaccine types: AstraZeneca (*n* = 350), Moderna Spikevax (*n* = 431), Pfizer-BioNTech Comirnaty (*n* = 379) and Sinopharm (*n* = 620).

### 2.2. Data and Specimen Collection

A structured questionnaire was used to collect data on demography, monthly income, history of previous SARS-CoV-2 infection, and COVID-19 vaccination status. Height and weight were measured using the free-standing stadiometer (Seca 217, Hamburg, Germany) and digital weighing scale (Camry-EB9063, China), respectively to calculate body mass index (BMI). Blood samples (10 mL) were collected from the participants at the time of enrollment. The study design is depicted in a flow chart showing details of sample collection, and analysis by various methods in each vaccination arm (Appendix A).

### 2.3. Specimen Processing

Plasma was separated from blood by centrifugation at 400× *g* and stored at −80 °C in a freezer. Peripheral blood mononuclear cells (PBMCs) were isolated (only from blood of SRGIH participants) by density gradient centrifugation at 500× *g* using Ficoll–Hypaque. The separated PBMCs were washed and then frozen in 10% di-methyl sulfoxide (DMSO) in fetal bovine serum (FBS) and preserved in liquid nitrogen until use.

### 2.4. Assessment of SARS-CoV-2-Specific Antibodies

Concentration of IgG antibodies directed to the SARS-CoV-2 spike (S) protein receptor binding domain (RBD) was determined in plasma by Elecsys^®^ Anti-SARS-CoV-2 S immunoassay (Roche Diagnostics GmbH, Mannheim). SARS-CoV-2 nucleocapsid (N)-specific antibodies (both IgG and IgM) were determined in plasma using an Elecsys^®^ Anti-SARS-CoV-2 N immunoassay (Roche), and categorized as seropositive and seronegative based on the antibody cut-off index (COI ≥ 1.0, reactive; COI < 1.0, non-reactive). Participants positive for N-specific antibodies in mRNA and adenovector vaccine groups were considered as individuals previously exposed to or infected with SARS-CoV-2. With inactivated whole-virus vaccine, it is not possible to differentiate whether N-specific antibody was generated due to vaccination or natural infection with SARS-CoV-2.

### 2.5. Pseudovirus Neutralization Assay (PNA)

Neutralizing potential of plasma antibodies against SARS-CoV-2 was determined using pseudovirus neutralization assay (PNA) in a subset of 40 participants (*n* = 10 from each vaccine group). Participants in mRNA and adenovector vaccine groups were selected based on N-specific antibody positivity and negativity (5 positives and 5 negatives from each group). From the inactivated vaccine group, participants with high (titers above 75% percentile) and low (titers below 25% percentile) S-antibody titers were selected (Appendix A). Pseudoviruses expressing the codon-optimized Wuhan reference strain’s spike protein and containing luciferase reporter were added sequentially to the heat-inactivated diluted plasma to prepare plasma–virus complexes. The complexes were then transferred onto plates, previously seeded with Vero E6 cells. Following incubation at 37° and 5% CO_2_, the luciferase substrate was added to the cells, and the luminescence intensity was measured on a luminescence plate reader. Obtained relative luminescence units (RLUs) were inversely proportional to the level of neutralizing antibodies present in the plasma. The reciprocal dilution of plasma required to inhibit pseudovirus infection by 50% (NT_50_) was used as a measure of neutralization potential of antibodies in each specimen.

### 2.6. Flow Cytometry of PBMC

Immune cell profiling of frozen PBMC was performed by flow cytometry in the same 40 participants as analyzed for the PNA assay. Cryopreserved PBMCs were thawed, washed, counted and stained using a cocktail of monoclonal antibodies conjugated to different fluorochromes (BD Biosciences, San Jose, CA, USA) for specific cell surface markers (Appendix A). At least 500,000 events per sample were acquired on a BD FACSCanto™ (BD Biosciences, San Jose, CA, USA). Acquired data were analyzed using FACS DIVA software (Tree Star, Inc., Ashland, OR, USA). The different cell populations were identified based on forward- and side-scatter characteristics and cell-specific surface receptors. At least 50,000 lymphocyte-gated cells were analyzed for T- and B-cell assessment. Appendix A shows the process of gating and obtaining profiles of immune cells.

### 2.7. ELISPOT Assay

Functional T-cell response was assessed in the PBMCs from 108 participants (27 participants from each vaccine group, selected following the same strategy as for PNA and FACS analysis) by determining SARS-CoV-2-peptide-specific interferon gamma (IFN-γ) secreting T cells using Human IFN-γ/IL-5 Double-Color ELISPOT kits (Cellular Technology Ltd., Shaker Height, OH, USA). PepMix™ SARS-CoV-2 (Spike Glycoprotein) was used as the stimulating antigen (JPT Peptide Technologies, Berlin, Germany).

### 2.8. Statistical Analysis Plan

Data are presented as geometric mean with standard deviation (SD) or number as a percent or median with interquartile range. To estimate the difference in antibody titers, neutralizing antibodies and IFN-γ-secreting T cells between the vaccine groups were analyzed with 2-way ANCOVA. ANCOVA was also used to determine the difference in S-specific antibody titers between RT-PCR and N-antibody-positive and -negative cases in each vaccine group. The model was adjusted for age, sex, BMI and family income. The difference was considered statistically significant when *p* < 0.05. Spearman correlation was used to assess the correlation between S-specific IgG titers, PNA titers and cell response. Statistical analysis was performed in STATA-15, and graphs were prepared in GraphPad Prism 8.3.1.

## 3. Results

### 3.1. Participants

About 52.0% of the participants were male. Post-vaccination symptoms were observed mostly in the Moderna- and Pfizer-vaccinated individuals after getting the first or second dose of vaccines. Common symptoms included fever, tiredness, muscle pain, headache and body ache. Out of 399 participants who went for RT-PCR testing, only 178 participants (54, 64, 39 and 21 in the AstraZeneca, Moderna, Pfizer and Sinopharm groups, respectively) tested positive for SARS-CoV-2 infection by RT-PCR.

### 3.2. Vaccine-Specific Antibody Response

Anti-S-IgG antibodies were highest in the participants vaccinated with Pfizer, followed by Moderna, AstraZeneca and Sinopharm vaccinees (Figure 1A). Participants previously infected with RT-PCR-confirmed SARS-CoV-2 before the first dose of the COVID-19 vaccine showed significantly higher post-vaccination antibody responses compared to uninfected individuals in the overall study population (beta(β) = 9547, 95% CI = 7268, 12,398) and in individual vaccine groups (Figure 1B). Similarly, participants seropositive for N-antigen showed higher anti-S-IgG antibody responses compared to those who were seronegative in the overall study population (β = 6772, 95% CI = 6412, 7009) and in Pfizer, Moderna and AstraZeneca vaccine groups (Figure 1C). Individuals who developed symptoms post-vaccination showed more anti-S-IgG antibodies compared to those who did not (Table 1).

### 3.3. Neutralizing Antibodies

The level of neutralizing antibodies (NT_50_) against pseudoviruses was significantly higher in Moderna- and Pfizer-vaccinated groups compared to the AstraZeneca and Sinopharm groups (Figure 2). Neutralization titers were found to be positively associated with anti-S antibodies (β = 1.29, 95% CI = 1.20, 1.38) and history of RT-PCR positivity (β = 2.57, 95% CI = 0.95, 6.92).

### 3.4. Immune Cell Profile

Participants vaccinated with AstraZeneca showed fewer plasmablasts (CD19^+^CD27^+^CD38^+^) and B-regulatory (B_reg_, CD19^+^CD24^++^CD27^+^) cells compared to the Pfizer and Moderna groups, respectively, but more memory B (B_MEM_, CD19^+^CD27^++^) cells compared to all three vaccine groups (Figure 3A). CD4^+^ effector (CD4^+^T_EFF_, CD45RA^+^CD45RO^−^CCR7^−^) cells were highest in the Pfizer group, while CD4^+^ naive (CD4^+^_N_, CD45RA^+^CD45RO^−^CCR7^+^) cells were highest in the Sinopharm group. Both these cell types were significantly less present in the AstraZeneca group compared to the Pfizer and Sinopharm group, respectively (Figure 3B). On the other hand, AstraZeneca vaccine recipients exhibited significantly more CD4^+^ T-central-memory (CD4^+^T_CM_, CD45RA-CD45RO^+^CCR7^+^) cells and CD8^+^ T cells compared to Pfizer and Sinopharm vaccinees, and more T-regulatory (T_reg_, CD4^+^CD25^++^CD127^low^) cells compared to Moderna and Sinopharm vaccinees. CD4^+^T_CM_ was also significantly higher in Moderna vaccine recipients compared to Pfizer and Sinopharm vaccinees (Figure 3B).

### 3.5. T Cell Function

Both Moderna (178.7 ± 1.29)- and Pfizer (200.0 ± 1.31)-vaccinated individuals exhibited significantly higher numbers of IFN-γ-secreting T cells/10^6^ PBMC compared to the AstraZeneca (36.8 ± 1.36) and Sinopharm groups (53.7 ± 1.32) (*p* < 0.001) (Figure 4). N-antibody-positive participants had higher numbers of IFN-γ-secreting T cells compared to N-antibody-negative individuals in the Pfizer (β = 0.64, 95% CI = 0.25, 1.02) and AstraZeneca (β = 0.64, 95% CI = 0.27, 1.01) groups. No such significant difference was noted in the Moderna group.

### 3.6. Correlation between S-Specific IgG Antibodies, NT_50_ and Cell Response

S-IgG titers were highly correlated with NT_50_ (r = 0.913, *p* = 0.000) and moderately correlated with Breg cells (r = 0.384, *p* = 0.008) and plasmablasts (r = 0.297, *p* = 0.045). The neutralizing antibody NT_50_ was also positively correlated with Breg cells (r = 0.325, *p* = 0.046) and plasmablasts (r = 0.379, *p* = 0.019). S-IgG and NT_50_ antibodies did not show any correlation with other cell types.

## 4. Discussion

Here, we report humoral and cellular immune responses in individuals vaccinated with four different COVID-19 vaccines . Moderna and Pfizer vaccine recipients showed significantly higher S-specific IgG antibody titers, neutralizing antibodies and IFN-γ-secreting T lymphocytes compared to AstraZeneca and Sinopharm vaccine recipients. On the other hand, memory B- and T-cell responses were highest in the AstraZeneca group. Participants previously infected with SARS-CoV-2 before the first dose of vaccines exhibited higher S-specific and neutralizing antibody titers compared to those who were uninfected.

We have demonstrated higher binding and neutralizing antibody responses in mRNA vaccine recipients compared to adenovector and killed whole-virus vaccine recipients. These findings are supported by several other reports [24,27,28,29,30]. A strong association between neutralizing antibodies and anti-S antibodies was also seen in this study as well as in other populations, including vaccine recipients and convalescent patients [24,29,31]. In invading host cells, the first and most crucial step for the virus is to bind to the angiotensin-converting enzyme 2 (ACE2) receptor of the host cell with the help of spike (S) glycoprotein. Pfizer, Moderna and AstraZeneca vaccines use this S protein as the immunogen with the aim of producing anti-S antibodies that neutralize the receptor-binding domain of the S protein [3,32,33]. While Pfizer and Moderna vaccines use codon-optimized sequences of mRNA for efficient expression of the full-length S protein delivered to the host cell through lipid nanoparticles (LNPs), the AstraZeneca vaccine utilizes DNA delivered through a chimpanzee adenovirus vector [34]. The rapid delivery of the mRNA into the host cell cytoplasm by LNP, direct translation into S protein, adjuvant properties of both mRNA and LNP and stabilizing mutations preventing the conformational change from the pre-fusion to the post-fusion structure of S protein [34,35,36] might explain the higher antibody response and neutralization capacity of the mRNA vaccines compared to adenovector vaccines. Again, formation of the post-fusion S and the concomitant dissociation of S1 due to inactivation and purification processes could compromise the antibody response of the Sinopharm vaccine [34]. The higher binding and neutralizing antibody response of mRNA vaccines may reflect the better protection offered by these vaccines compared to inactivated- and vector-based vaccines [37,38]. Elevated antibodies in participants previously exposed to or infected with SARS-CoV-2 as supported by earlier studies would indicate the importance of the memory B-cell and probably also T-cell response in providing a substantial boost to the humoral response [39,40].

Rapidly expanding plasmablasts are the key to successful humoral immunity as they produce high antibody titers in the initial phase of an infection or vaccination [41]. The lower number of plasmablasts in the AstraZeneca vaccine recipients is aligned with the lower production of antigen-specific antibodies compared to mRNA vaccine recipients. On the other hand, the highest frequency of CD19^+^CD27^++^ B_MEM_ cells that produce enormous numbers of antibodies in response to re-infection may reflect longevity of immune memory in the AstraZeneca group [42]. The longer duration of immunological memory in AstraZeneca vaccine recipients was also reflected by the highest number of CD4^+^T_CM_ cells. Central memory T cells migrate to secondary lymphoid organs, where they proliferate and differentiate into effector T cells upon exposure to antigenic stimulation [43]. Previous studies have shown that transcriptionally active adenovirus vector genomes can persist for months after inoculation, leading to continuous production of low amounts of antigen, which has been proposed to contribute to expanding and maintaining memory cell populations [44,45].

IFN-γ, a key moderator of cell-mediated immunity that is expressed by activated CD4^+^ and CD8^+^ T cells, can mediate direct antiviral function and enhance the antiviral effects of CD8^+^ T cells [46,47,48]. Even though the proportion of total cytotoxic T cells was higher in the AstraZeneca vaccine recipients, lower numbers of IFN-γ-secreting T cells in parallel to the reduced proportion of CD4+ T_EFF_ cells may indicate the somewhat diminished capacity of the effector function of the cellular immune arm compared to mRNA vaccines.

Treg may protect the host from acute viral infections by downregulating immunopathogenic mechanisms of tissue damage [49,50]. Maintenance of immune homeostasis through suppression of inflammatory responses is also evident for Breg cells [51]. SARS-CoV-2 infection induces inflammatory pathogenesis during COVID-19 onset by downregulating both Treg and Breg cells, especially in severe and critical patients [52,53]. A higher number of CD19^+^CD24^++^CD27^+^ Breg cells after administration of mRNA-based vaccines and higher numbers of CD4^+^CD25^++^CD127^low^ Treg cells in AstraZeneca vaccine recipients suggest that these two types of vaccines use different immunoregulatory pathways. Moderna shots are known to produce more adverse effects than other vaccines, and this could be partially explained by the lower frequency of Treg cells in these vaccine recipients. In participants receiving the AstraZeneca vaccine, a higher number of Treg cells might control the differentiation of CD4^+^ cells into the effector phenotype [54]. In fact, a favorable ratio of effector to regulatory cells is needed to strike a balance in the immune response. S-specific antibody titers as well as NT50 antibodies correlated with plasmablasts and Breg cells, further highlighting the roles of B-cell lineage in producing functional antibodies against SARS-CoV-2 among the vaccinees.

Individuals vaccinated with inactivated vaccines had a lower number of memory B cells, CD4+ T-memory cells, CD8+ cells and Treg cells. Insufficient cellular responses along with lower titers of S-specific and neutralization antibodies suggest lower protection as well as a shorter duration of immunological memory offered by inactivated vaccine.

Limitations of this study include not collecting blood specimens before vaccination as well as before infection in the individuals with PCR-confirmed SARS-CoV-2 infections. However, we collected plasma from individuals within 2–4 weeks after the second dose of vaccination to obtain peak antibody responses to SARS-CoV-2 antigens; the timing was uniform across all vaccine groups. One limitation was that we did not determine the antigen-specific phenotypic characterization of peripheral immune cells. Another limitation was the small sample size of the participants in the PNA, IFN-γ-secreting T-cell and phenotypic cell assessment components due to budget constraints. However, all the results point to the same direction, supporting the antibody response findings and further strengthening the data.

## 5. Conclusions

In conclusion, the serological data from recipients vaccinated with the four vaccines suggest that mRNA-vaccinated individuals had higher humoral immunity compared to adenovector and inactivated vaccine recipients. However, total CD8^+^T cells, pools of central memory CD4^+^T cells and memory B cells were higher in adenovector vaccine recipients. Taken together, the findings of the study underscore the need for monitoring of both arms of the immune system after vaccination. Long-term follow-up of vaccinated individuals will show comparative data on sustaining or waning of humoral and cellular immune responses generated from the different vaccination groups. Moreover, booster (third) dosing with the heterologous vaccine in the same population would provide valuable information on whether the combination of different humoral and cellular immune responses evoked by different vaccines leads to higher and prolonged immunity compared to homologous vaccines.

## Figures and Tables

**Figure 1 vaccines-10-01498-f001:**
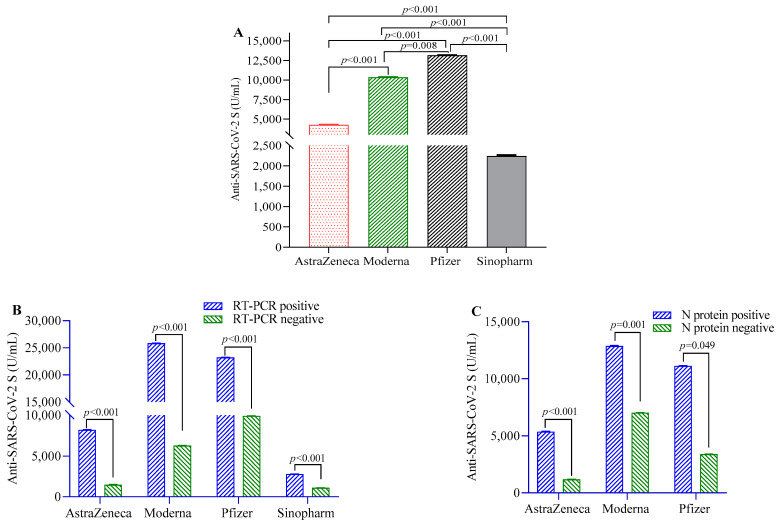
SARS-CoV-2 S-protein-specific antibody levels in study participants after two doses of vaccination. (**A**) Comparison of antibody levels between different SARS-CoV-2 vaccines; (**B**) differences in antibody titer between RT-PCR positive and negative participants; (**C**) comparison of antibody levels between participants positive and negative for SARS-CoV-2 N-protein-specific antibody. A 2-way ANCOVA model was used to estimate the *p*-value, and the model was adjusted for age, sex, income and BMI.

**Figure 2 vaccines-10-01498-f002:**
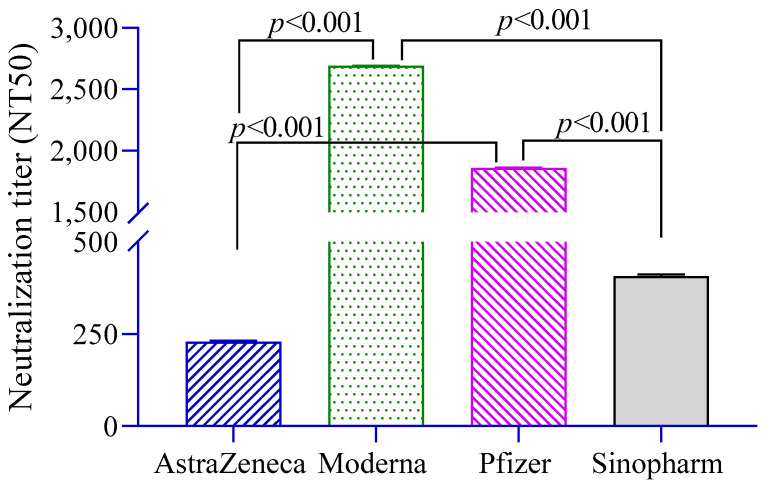
Neutralizing antibody titers in different vaccine groups. A 2-way ANCOVA was used to estimate the *p*-value, and the model was adjusted for age, sex, income and BMI.

**Figure 3 vaccines-10-01498-f003:**
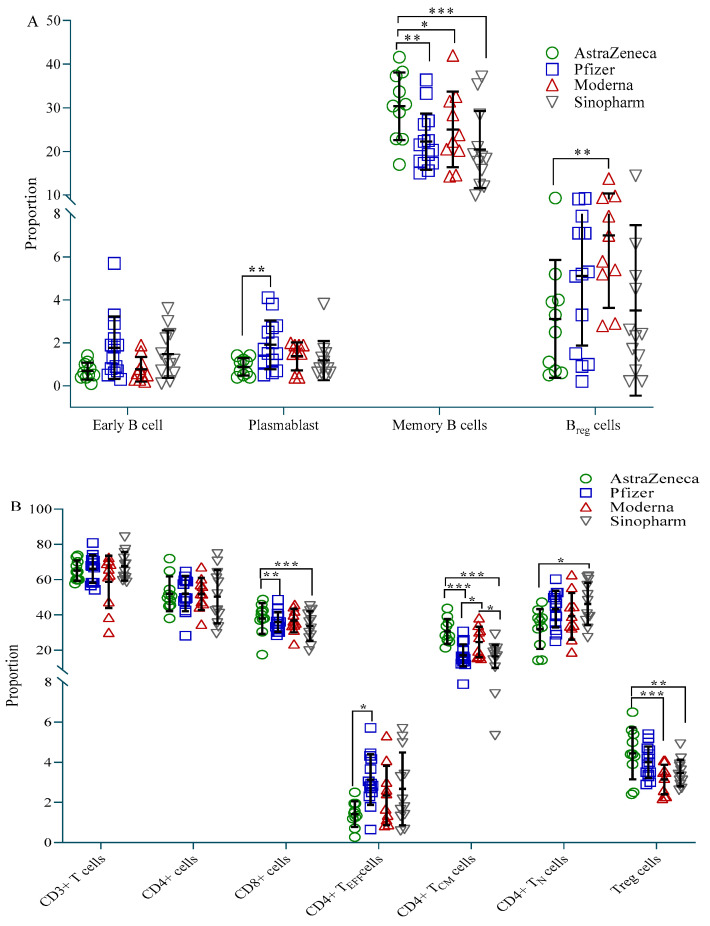
Proportion of immune cells in different vaccine groups. (**A**) Proportion of B cell subtypes; (**B**) Proportion of T cell subtypes. A 2-way ANCOVA was used to estimate the *p*-value, and the model was adjusted for age, sex, income and BMI. * *p* < 0.05, ** *p* < 0.01, *** *p* < 0.001.

**Figure 4 vaccines-10-01498-f004:**
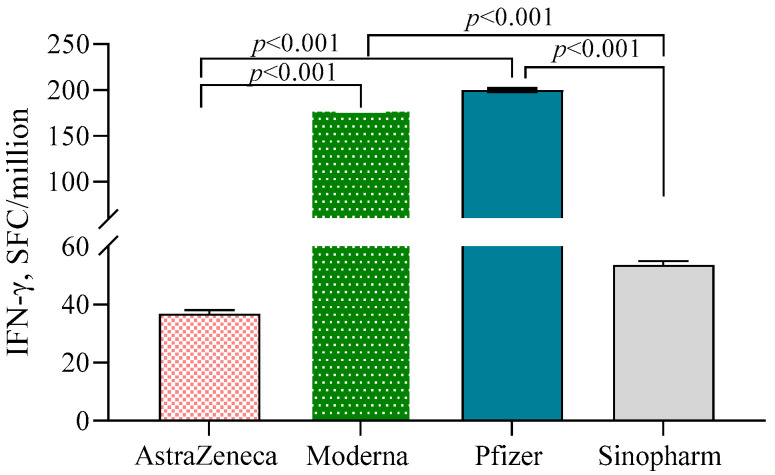
IFN-γ-secreting T cells in different vaccine groups. A 2-way ANCOVA was used to estimate the *p*-value, and the model was adjusted for age, sex, income and BMI.

**Table 1 vaccines-10-01498-t001:** Differences in antibody titers between participants with and without COVID-like symptoms after getting 1st and 2nd doses of COVID-19 vaccines.

Symptoms	After 1st Vaccination		After 2nd Vaccination	
	β-Coff (95% CI)	*p*-Value	β-Coff (95% CI)	*p*-Value
Fever	2.69 (1.86, 3.80)	<0.001	2.88 (2.04, 4.07)	<0.001
Tiredness	1.91 (1.23, 2.98)	0.004	1.86 (1.17, 2.95)	0.008
Muscle pain	1.70 (1.17, 2.45)	0.006	1.29 (0.91, 1.82)	0.149
Headache	2.04 (1.20, 3.47)	0.009	2.75 (1.62, 4.79)	<0.001
Whole body pain	1.17 (0.85, 1.62)	0.332	1.27 (0.91, 1.78)	0.159

Multivariate regression model was used to calculate the mean difference (β-coefficient) between participants having each symptom and asymptomatic participants and to estimate the *p*-value. Log-transformed data were used for the calculation. The regression model was adjusted by age, sex, history of SARS-CoV-2 infection (RT-PCR confirmed) and household income.

## Data Availability

Anonymous participant data and a data dictionary for each variable analyzed in this article, as well as the study protocol, the statistical analysis plan and the informed consent form will be made available when the study is complete, upon requests directed to the corresponding author (rubhana@icddrb.org). Data can be shared after approval of a proposal through a secure online platform. The institutional review board at the icddr,b will review the proposal and will approve. Additionally, data sharing will depend on the published data access rules of the icddr,b. A standard 385 data access agreement by the icddr,b will need to be signed.

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
