# Peer review of "Comparison of the Immune Responses to COVID-19 Vaccines in Bangladeshi Population"

_vaccines, 2022, doi:10.3390/vaccines10091498_

Round 1

Reviewer 1 Report

In this work, the authors compare immune response profiles among people in Bangladesh given 4 different COVID-19 vaccines. The approach is interesting in terms of comparing the potency of the vaccines, but the methodology and results are presented in a way making it hard to understand exactly what has been done. It is mentioned that 1780 participants were enrolled, but not how many for each vaccine. Also, it is mentioned that the enrolment took place from November 2021 to May 2022, and that participants were enrolled 4-8 weeks after receiving two vaccine doses, but it is not clear whether this time gap was similar for all 4 vaccines.  Apparently, some of the participants got infected with SARS-CoV-2 virus before, between or after vaccination and these had higher antibody titers. But again, the results are hard to interpret as it is not specified when the individuals got infected (in relation to time of vaccination/blood sampling), whether they experienced clinical disease, and which vaccine they received. In line 94 (p2 bottom) it is stated that vaccinees positive for N-specific antibodies were considered positive for infection, but probably the authors mean positive for having encountered the infection at a certain point? Only participants being positive for virus by Q-PCR or antigen test at the time of sampling should be considered positive for infection (at the time of blood sampling). It is not clear whether this was examined?

Samples from 40 participants were selected for testing for neutralizing antibodies and Immune cell profiling and it is mentioned that for the Sinopharm group individuals from the 75% higher and 25% lower fractiles were selected, but otherwise info on how the individuals were selected and how many from each group is not clear either. The immune cell profiling was done by flow cytometry on cells, that had been frozen. Frozen cells/dead cells are not optimal for flow cytometry and the results (Fig 3) indicate only marginal differences between the groups. Infor on how the cells were preserved is missing.

The results in Table 1 are hard to understand. The reference, i.e. people without any side effects of vaccination is missing and it is not clear how the data were collected (e.g. based on interviews, patient journals or others?).

In general, data are presented in a rather summaric way making it difficult for the reader to determine whether the interpretations/authors conclusions appear justified.
It is necessary to include detailed supplementary data showing how and which samples were selected for the different analysis, whether and when the participant had SARS-CoV-2 infection including severity of disease, raw data from the various analyses Including flow cytometry, etc.

Author Response

Comments and Suggestions for Authors

In this work, the authors compare immune response profiles among people in Bangladesh given 4 different COVID-19 vaccines. The approach is interesting in terms of comparing the potency of the vaccines, but the methodology and results are presented in a way making it hard to understand exactly what has been done.

It is mentioned that 1780 participants were enrolled, but not how many for each vaccine.

Response: We thank the reviewer for indicating the important omission. We have now included the number of participants receiving each vaccine in the ‘Materials and Methods’ section (lines 86-88) and in supplementary Figure 1.

Also, it is mentioned that the enrolment took place from November 2021 to May 2022, and that participants were enrolled 4-8 weeks after receiving two vaccine doses, but it is not clear whether this time gap was similar for all 4 vaccines. 

Response: The participants were actually enrolled 2-4 weeks after receiving two vaccine doses, and this time gap was similar for all 4 vaccines. The time gap has been corrected in the manuscript in ‘Materials and Methods’ section (line 83).

Apparently, some of the participants got infected with SARS-CoV-2 virus before, between or after vaccination and these had higher antibody titers. But again, the results are hard to interpret as it is not specified when the individuals got infected (in relation to time of vaccination/blood sampling), whether they experienced clinical disease, and which vaccine they received.

Response: We have revised the sentences in the ‘Results’ section (lines 176-177) as well as in the ‘Discussion’ section (lines 240-242) to state that the participants were previously infected with SARS-CoV-2 only before the 1st dose, not between the two doses of vaccines. The information on exactly when (how many days before the 1st dose) the participants got infected could not be ascertained for many participants. Only those who could show RT-PCR results were confirmatory cases. Some of the participants could clearly mention the time (month) of onset of COVID-like symptoms while others could vaguely remember. Majority of the SARS-CoV-2 infected participants had mild or no symptoms, thus did not require any treatment. Therefore, we did not look for the association between clinical severity and antibody response.

Out of 399 participants, only 178 participants tested positive for SARS-CoV-2 infection by RT-PCR. Of them 54 belonged to AstraZeneca group, 64 Moderna, 39 Pfizer and 21 to Sinopharm group (lines 171-173). We have also revised Figure 1B to show the difference in antibody titers between RT-PCR positive and negative cases among different vaccine recipients.

In line 94 (p2 bottom) it is stated that vaccinees positive for N-specific antibodies were considered positive for infection, but probably the authors mean positive for having encountered the infection at a certain point? Only participants being positive for virus by Q-PCR or antigen test at the time of sampling should be considered positive for infection (at the time of blood sampling). It is not clear whether this was examined?

Response: As per suggestion of the reviewer, we have now rephrased the sentence in the ‘Materials and Methods’ section (lines 110-112) to state that the vaccinees positive for N-specific antibodies reflected previous exposure to/infection with SARS-CoV-2 virus. None of the participants had any symptoms at the time of blood collection and were not tested by RT-PCR for SARS-CoV-2.

Samples from 40 participants were selected for testing for neutralizing antibodies and Immune cell profiling and it is mentioned that for the Sinopharm group individuals from the 75% higher and 25% lower fractiles were selected, but otherwise info on how the individuals were selected and how many from each group is not clear either.

Response: In a subset of 40 participants, 10 from each vaccine group were selected for testing of neutralizing capacity of antibodies. Selection of participants in m-RNA and adenovirus vaccine groups were based on N-specific antibody positivity (5 positives and 5 negatives in each group) to include participants who had previously encountered SARS-CoV-2 infection and those who did not. Another selection criterion was the availability of participants who were negative for N-specific antibody in each vaccine group. Majority of the participants were N-antibody positives. In one vaccine group (Pfizer), only 20 participants were negative for N-antibody. We, therefore selected 13 N-antibody negative participants for ELISPOT assay and 5 for flowcytometry from each group; similar numbers of N-antibody positive participants were included in the analysis. Since, Sinopharm vaccine is a whole cell vaccine, it is not possible to determine whether N-specific antibody was generated due to vaccination or natural infection with SARS-CoV-2, and therefore individuals with high and low S-antibody titers were selected instead of N-specific antibodies. Relevant changes have now been made in the ‘Materials and Methods’ section (lines 118-123) and in Supplementary Figure 1.

The immune cell profiling was done by flow cytometry on cells, that had been frozen. Frozen cells/dead cells are not optimal for flow cytometry and the results (Fig 3) indicate only marginal differences between the groups. Infor on how the cells were preserved is missing.

Response: We agree with the reviewer that fresh cells are the best choice for analysis of cellular immunity including flow-cytometry. However, using fresh cells are not always feasible when collected from remote areas; many researchers perform flow cytometry in frozen cells to run in batches. Since, we had to select the samples based on antibody titers, we used frozen cells and perform the analysis batch-wise. Information on how the cells were preserved in liquid nitrogen has been given in the ‘Materials and Methods’ section ( lines 101-102).

The results in Table 1 are hard to understand. The reference, i.e. people without any side effects of vaccination is missing and it is not clear how the data were collected (e.g. based on interviews, patient journals or others?).

Response: The results in the Table 1 are given as β-Coefficient, i.e. mean difference in antibody titers between participants having each symptom and asymptomatic participants (reference group). The title and footnote of the table have now been revised to clarify the results (page 5). Data supporting Table 1 have been provided below (following the responses below).We have used a structured questionnaire to collect data on demography, monthly income, approximate time of previous infection with SARS-CoV-2 (history of previous infection), COVID-19 vaccination dates. Data collection method has now been included in the ‘Materials and Methods’ section (lines 91-92).

In general, data are presented in a rather summaric way making it difficult for the reader to determine whether the interpretations/authors conclusions appear justified. It is necessary to include detailed supplementary data showing how and which samples were selected for the different analysis, whether and when the participant had SARS-CoV-2 infection including severity of disease, raw data from the various analyses including flow cytometry, etc.

Response: We have added a flow chart in the supplementary Figure 1 showing time of sample collection, and samples being used for different analysis. We have also clarified how the samples were selected for different analysis.

In response to the earlier comment, we have already mentioned how many participants had RT-PCR confirmed SARS-CoV-2 infection, when they got infected, and how many infected participants belonged to each vaccine group. Since most of the SARS-CoV-2 infected participants had mild or no symptoms, severity of COVID-19 disease was thus not taken into consideration for analysis.

            As mentioned in data availability statement (page 10), anonymous participant data and a data dictionary for each variable analyzed in this article, can only be made available when the study is complete, and requests are directed to the corresponding author. Our study is ongoing. Moreover, there will be a need to sign a standard 385 data access agreement by the icddr,b. Therefore, it may not be possible to share data in supplementary file. However, we are sharing some data supporting our results here for reviewer’s assessment.

Data for Figure 1

SARS-CoV-2 S-protein specific antibody levels in study participants after two doses of vaccination.

Vaccine Types

Anti SARS CoV-2 S (U/mL)

AstraZeneca (n=350)

4275.6±20.0

Moderna (n=431)

10351±22.0

Pfizer (n=379)

13152±21.0

Sinopharm (n=620)

2238.7±26.0

Data presented as Geometric mean ± SD

SARS-CoV-2 S-protein specific antibody levels in RT-PCR positive or negative participants (n=399)

Vaccine Types

RT-PCR positive (n=178)

RT-PCR negative (221)

p-value

AstraZeneca (n=54 & 78)

8222±27.1

1486.3±28.4

<0.001

Moderna (n=64 & 56)

25858±19.4

6301±20.9

<0.001

Pfizer (n= 39 & 32)

23243±12.7

9919±12.1

<0.001

Sinopharm (n=21 & 55)

2800±19.4

1109.4±11.6

<0.001

Overall

13728±49.5

2623±52.0

<0.001

Data presented as Geometric mean ± SD. A total of 399 participants got tested for COVID-19 by RT-PCR.

Data for Table 1

S-antibody titers among participants with or without COVID-like symptoms after getting 1st and 2nd doses of COVID-19 vaccines.

Symptoms

After 1st vaccination

After 2nd vaccination

Presence of symptoms

Absence of symptoms

p-value

Presence of symptoms

Absence of symptoms

p-value

Fever

12023±3.31

3467.4±5.01

<0.001

11749±3.72

3311.3±4.79

<0.001

Tiredness

9772±3.24

4168.7±5.13

0.004

9550±4.27

4265.8±5.01

0.008

Muscle pain

5128.6±4.57

3467.4±6.46

0.006

4897.8±4.79

4265.8±5.50

0.149

Headache

11220±2.88

4265.8±5.13

0.009

13490±2.63

4265.8±5.05

<0.001

Whole body pain

4786.3±4.47

4677.4±5.37

0.332

5128.6±4.57

4570.9±5.25

0.159

Data presented as Geometric mean ± SD

Data for Figure 2

Vaccine Types

Neutralization titer (NT50)

AstraZeneca (n=10)

229.1±2.88

Moderna (n=10)

2692.0±1.95

Pfizer (n=10)

1862.0±1.78

Sinopharm (n=10)

407.4±4.40

Neutralizing antibody titers in different vaccine groups.

Data presented as Geometric mean ± SD

Data for Figure 3

Proportion of T and B cell subtypes in different vaccine groups

Immune cells

AstraZeneca (n=10)

Moderna (n=10)

Pfizer (n=10)

Sinopharm (n=10)

CD3+ T cells

69.8±2.73

67.1±2.73

63.6±2.89

66.2±2.65

CD4+ cells

43.2±3.41

43.6±3.40

54.6±3.61

57.1±3.30

CD8+ cells

35.5±1.76

33.4±1.76

34.2±1.87

28.4±1.71

Treg cells

4.38±0.31

2.82±0.31

1.18±0.33

1.08±0.30

CD4+ TCM cells

32.9±2.39

25.6±2.39

18.2±2.53

20.1±2.32

CD4+ TEFF cells

1.30±0.39

2.05±0.39

3.18±0.42

1.66±0.38

CD4+ TN cells

34.8±3.74

39.0±3.73

40.6±3.96

38.1±3.62

Breg cells

3.90±1.14

8.68±1.14

5.58±1.21

3.69±1.10

Early B cell

0.70±0.26

0.85±0.25

1.08±0.27

1.00±0.24

Plasmablast

2.87±0.50

3.21±0.50

1.33±0.53

0.61±0.49

Memory B cells

31.2±2.15

23.7±2.14

15.7±2.28

10.5±2.08

Data presented as mean ± SD

Data for Figure 4

IFN-g secreting T cells in different vaccine groups

Vaccine type

IFN-γ

AstraZeneca (n=27)

36.8±1.36

Moderna (n=27)

178.7±1.29

Pfizer (n=27)

200.0±1.31

Sinopharm (n=27)

53.7±1.32

Data presented as geometric mean ± SD

As suggested by the reviewer, we have tried to improve the overall content of the manuscript beside responding to the specific comments.

Reviewer 2 Report

The manuscript compares the response of the individuals to different vaccines. In general, the manuscript is well presented. The analysis of the antibodies and cellular response are important. The authors, however, did not analyze CD8 antiviral response, which should have been important. The role of NK, NK and T gamma delta cells could also be important in response to the vaccine since there is a comparison of different types and should be discussed at the end of the manuscript. The authors also should analyze whether there is a statistical relationship or not, between antibodies, neutralizing antibodies and cellular response. Finally, in the opinion of the authors is the inactivated vaccine less protective? and why?  

Author Response

Comments and Suggestions for Authors

The manuscript compares the response of the individuals to different vaccines. In general, the manuscript is well presented. The analysis of the antibodies and cellular response are important. The authors, however, did not analyze CD8 antiviral response, which should have been important. The role of NK, NK and T gamma delta cells could also be important in response to the vaccine since there is a comparison of different types and should be discussed at the end of the manuscript.

Response: We very much appreciate the reviewer’s suggestion to include the role of different cell types as well as antiviral response of CD8 cells. However, studying these cell types are out of the scope of this short communication. We hope to consider these analyses in the future work.

The authors also should analyze whether there is a statistical relationship or not, between antibodies, neutralizing antibodies and cellular response.

Response: We thank the Reviewer for an excellent suggestion. Spearman’s correlation analysis showed that S-IgG titers were highly correlated with NT50 antibodies (r=0.913, p=0.000) indicating that the spike peptide specific antibodies closely reflected neutralizing capacities of these antibodies. S-IgG titers as well as NT50 antibodies correlated with Breg cells and plasmablasts further emphasizing the roles of B cell lineage in producing functional antibodies against SARS-CoV-2 among the vaccine recipients. We have included a separate subsection 3.6 in ‘Results section. to describe the correlation between antibodies and neutralizing antibodies and cellular response (lines 228-232).

Finally, in the opinion of the authors is the inactivated vaccine less protective? and why?  

Response: Referring to a systemic review and meta-analysis of different vaccines at phase 3 by Fan Y-J 2021 (PMID 34579226), we have discussed that lower binding and neutralizing antibody response of inactivated vaccine (as well as AstraZeneca vaccine) may offer lower protection compared to mRNA -based vaccines (lines 262-264). We further found that individuals vaccinated with inactivated vaccine had lower number of memory B cells, CD4+ T memory cells, CD8+ cells and Treg cells. In the case of AstraZeneca vaccine, even though antibody response was lower compared to that generated by mRNA vaccines, the memory B cell and CD4+ T central memory cell responses were higher. Altogether, these findings indicate lower protection as well as shorter duration of immunological memory offered by inactivated vaccine. These findings have now been discussed in page 9 (lines 303-306).

As suggested by the reviewer, we have tried to improve the overall content of the manuscript beside responding to the specific comments.
